# The Regulatory Roles of Mitochondrial Calcium and the Mitochondrial Calcium Uniporter in Tumor Cells

**DOI:** 10.3390/ijms23126667

**Published:** 2022-06-15

**Authors:** Linlin Zhang, Jingyi Qi, Xu Zhang, Xiya Zhao, Peng An, Yongting Luo, Junjie Luo

**Affiliations:** 1Beijing Advanced Innovation Center for Food Nutrition and Human Health, College of Food Science and Nutritional Engineering, China Agricultural University, Beijing 100083, China; zll900914@163.com; 2Key Laboratory of Precision Nutrition and Food Quality, Department of Nutrition and Health, China Agricultural University, Beijing 100193, China; qipeiyan2992@163.com (J.Q.); zhangx94@cau.edu.cn (X.Z.); 13051239388@163.com (X.Z.)

**Keywords:** mitochondrial calcium, calcium homeostasis, calcium regulation, MCU, tumor

## Abstract

Mitochondria, as the main site of cellular energy metabolism and the generation of oxygen free radicals, are the key switch for mitochondria-mediated endogenous apoptosis. Ca^2+^ is not only an important messenger for cell proliferation, but it is also an indispensable signal for cell death. Ca^2+^ participates in and plays a crucial role in the energy metabolism, physiology, and pathology of mitochondria. Mitochondria control the uptake and release of Ca^2+^ through channels/transporters, such as the mitochondrial calcium uniporter (MCU), and influence the concentration of Ca^2+^ in both mitochondria and cytoplasm, thereby regulating cellular Ca^2+^ homeostasis. Mitochondrial Ca^2+^ transport-related processes are involved in important biological processes of tumor cells including proliferation, metabolism, and apoptosis. In particular, MCU and its regulatory proteins represent a new era in the study of MCU-mediated mitochondrial Ca^2+^ homeostasis in tumors. Through an in-depth analysis of the close correlation between mitochondrial Ca^2+^ and energy metabolism, autophagy, and apoptosis of tumor cells, we can provide a valuable reference for further understanding of how mitochondrial Ca^2+^ regulation helps diagnosis and therapy.

## 1. Introduction

Mitochondria are involved in a series of cellular biological processes such as adenosine triphosphate (ATP) generation, apoptosis, and cell cycle regulation to maintain the cell’s life activities [1]. Calcium ions (Ca^2+^) are distributed in the mitochondrial intermembrane gap and matrix [2]. Ca^2+^ shuttles between mitochondria and cytoplasm through different transport mechanisms, regulating the life activities of mitochondria and even the whole cell. Ca^2+^ is an indispensable messenger for many important physiologic processes, including metabolism, cell proliferation and death, protein phosphorylation, gene transcription, neurotransmission, contraction, and secretion [3]. The level of intracellular Ca^2+^ depends on the release of endoplasmic reticulum (ER) Ca^2+^ and the inflow of extracellular Ca^2+^ [4]. In animal body fluids and tissues, the concentration of Ca^2+^ varies between 2.1 and 2.6 mM [5] and the unit of total Ca^2+^ concentration in cells is also mM. However, in the cytoplasm of most cells, the concentration of free Ca^2+^ is about 10,000 times lower. In cells, inorganic compounds and low molecular weight organic molecules usually bind Ca^2+^ with low affinity and will not reduce their free concentration to nM, which is necessary for Ca^2+^ to effectively perform their signaling functions [6]. Abnormal Ca^2+^ homeostasis is one of the common pathological mechanisms of many diseases. Studies have shown that Ca^2+^ can not only be absorbed and released by mitochondria, but also the process of Ca^2+^ uptake and release by mitochondria plays an important role in maintaining cytoplasmic calcium homeostasis [7,8,9,10,11].

Ca^2+^ plays an indispensable role in signal transduction from cell surface receptors to the cytoplasm and from the cytoplasm to mitochondria, so as to jointly regulate cell metabolism [12]. Cytoplasmic calcium oscillation is the most prominent signal in cells, which refers to the transmission of a variety of regulatory information by cytosolic Ca^2+^ ([Ca^2+^]c) in the form of concentration oscillation [13]. Inositol 1,4,5-trisphosphate (IP_3_)-induced intracellular Ca^2+^ mobilization results in an increase in mitochondrial Ca^2+^ ([Ca^2+^]m) [14]. IP_3_-dependent hormone-induced [Ca^2+^]c oscillation is effectively transmitted to mitochondria in the form of [Ca^2+^]m oscillation [15]. Moreover, it has been reported that the concentration of free Ca^2+^ in mitochondria is closely related to the level of energy metabolism and the change in membrane permeability [16]. Mitochondrial Ca^2+^ accumulation triggers the activation of the mitochondrial metabolic mechanism, which increases ATP synthesis in the mitochondria and the ATP level in cytoplasm [17]. The uptake and release of mitochondrial Ca^2+^ also affects the intracellular calcium signal [18]. The abnormality in these calcium signaling-related activities is significantly related to the occurrence and development of heart disease, epilepsy, and neurodegenerative diseases [19].

At present, the incidence and mortality rate of malignant tumors is increasing year by year and it is the primary cause of death from all kinds of diseases. It is estimated that by 2040, 28.4 million new cases of cancer will be diagnosed worldwide, which represents an increase of 47% since 2020 [20]. There’s ample evidence that Ca^2+^ signaling is a key regulator in a series of tumor cell processes, including tumor growth, progression, and metastasis [21]. The alteration of Ca^2+^ is a hallmark of many tumors. For instance, Ca^2+^ is decreased in pancreatic cancer, colon cancer, and prostate cancer, while Ca^2+^ is increased in breast cancer and hepatocellular carcinoma (HCC) [22]. Altered Ca^2+^ signaling accelerates lipid accumulation and may promote HCC development [23]. Mitochondrial Ca^2+^ uptake is necessary for the progression of triple-negative breast cancer (TNBC) in vivo and it can also activate the hypoxia-inducible factor-1 alpha (HIF-1α) signal pathway, which contributes to tumor growth and metastasis [24]. In addition, intercellular Ca^2+^ signaling is altered in urinary bladder carcinoma cells [25]. Orai1-store-operated Ca^2+^ entry (SOCE) intracellular Ca^2+^ oscillation upregulation can activate downstream pathways, stimulate the proliferation and migration of esophageal squamous cell carcinoma (ESCC) cells, enhance their ability to invade other tissues, and promote the formation and growth of ESCC tumors in vitro and in vivo [26]. Meanwhile, SOCE also contributes to melanoma progression [27].

Regulated elevations in Ca^2+^ are required for the activity of several mitochondrial enzymes and this, in turn, regulates mitochondria-derived reactive oxygen species (ROS) generation; this is a known driver of pro-tumorigenic redox signaling, resulting in the activation of pathways implicated in cellular proliferation, metabolic alterations, stress adaptations and cell death [28,29,30,31,32]. Numerous studies have demonstrated that mitochondrial Ca^2+^ homoeostasis is involved in the metabolism, apoptosis, proliferation and other important processes of tumor cells [33,34]. In this review, we outline the role of mitochondrial Ca^2+^ in the regulation of tumor cell development and its molecular mechanisms, which is conducive to providing a basis for tumor therapy via targeting mitochondrial Ca^2+^ homoeostasis and regulation.

## 2. Regulation of Mitochondrial Ca^2+^

Due to the outer membrane of mitochondria possessing a high permeability for Ca^2+^, the concentration of Ca^2+^ in the membrane gap is equivalent to that in the cytoplasm [35]. In the resting state of cells, the concentration of Ca^2+^ in cytoplasm is about 100nM. When the cells are excited, the concentration of Ca^2+^ in the cytoplasm can rise to 1–3 µM. [36]. In fact, the uptake and release of Ca^2+^ by mitochondria can be regulated by the one-way transport mechanism or transporter [37]. The mitochondrial Ca^2+^ influx is mainly mediated by the mitochondrial calcium uniporter (MCU), voltage-dependent anion-selective channel (VDAC), and mitochondrial ryanodine receptor transporter. Furthermore, the mitochondrial Ca^2+^ efflux pathways mainly include leucine zipper/EF hand-containing transmembrane-1 (LETM1), mitochondrial Na^+^/Ca^2+^ exchanger (NCLX), and mitochondrial permeability transition pore (MPTP).

MCU is a Ca^2+^ channel ubiquitous in mitochondrial intima [38]. It is generally considered to be a key Ca^2+^ transporter [39] and silencing the MCU can severely abrogate mitochondrial Ca^2+^ uptake [40] (Figure 1). Knockout of the MCU completely inhibited mitochondrial Ca^2+^ uptake triggered by several stimuli in different cell types [41]. The MCU is an ion channel with electrophysiological characteristics. Ca^2+^ uptake through the MCU is driven by an electrochemical gradient. The MCU and related regulating molecules, including the essential MCU regulator (EMRE), mitochondrial calcium uptake (MICU)1, MICU2, MICU3, MCU-dominant negative beta subunit (MCUb), and MCU regulator 1 (MCUR1), form a large complex to manipulate the activities of the MCU [42]. The changes in the expression of these regulators are different in different cancer cells. For example, in pancreatic cancer cells, MICU1 and MICU2 are increased, while EMRE is decreased [43]. In breast cancer cells, MCU is elevated but MCUb is reduced [44]. In ovarian cancer cells, MICU1 mRNA is enhanced [45]. In HCC cells, the MCU, MCUR1, and MICU2 are elevated, while MICU1 is in decline [46].

In higher eukaryotes, the EMRE mediates MICU1/MICU2 to regulate Ca^2+^ transport through a leverage mechanism. MICU1/MICU2 is associated with the MCU through the EMRE. Each MICU1 interacts with two EMRE subunits. The interaction sites are located at the N-terminal poly K, s339k340k341 domain of MICU1 and the C-terminal of the EMRE [47]. The regulation of MCU activity by MICU1 and MICU2 involves a gating mechanism: when cells in a resting state and the concentration of intracellular Ca^2+^ is low, MICU1−MICU2 inhibits Ca^2+^ from entering the mitochondria through the MCU. When cells are stimulated by signals and the concentration of Ca^2+^ in the cytoplasm increases and exceeds a certain threshold (more than about 1 mM), MICU1−MICU2 allows Ca^2+^ to enter the mitochondria through the MCU [48]. Down regulation of MICU1 can reduce Ca^2+^ flux, decrease mitochondrial oxidative phosphorylation (OXPHOS) and ATP production, and activate AMPK-dependent autophagy [49]. In parallel, MICU1 also regulates the cristae junction to maintain the structural mitochondrial membrane framework, and without the cristae junction, it can mediate uncoupling and increase ROS production [50,51].

MICU1 is upregulated in ovarian cancer cells and its expression is closely related to the survival of cancer cells and tumor growth [52]. In this pathway, MICU1 induces the accumulation of mitochondrial Ca^2+^ and the production of ROS, suggesting that the binding of MICU1 to the MCU is necessary for the function of the MCU complex and the entry of Ca^2+^ into mitochondria is a prerequisite survival factor of cancer cells. MICU1 has been shown to be methylated by protein arginine methyltransferase 1 (PRMT1) in cancer cells, yielding decreased Ca^2+^ sensitivity and reduced Ca^2+^ entry. UCP2/3 is fundamental for mitochondrial Ca^2+^ uptake in cancer cells [53]. When it binds to methylated MICU1, it can normalize the Ca^2+^ sensitivity of MICU1 and re-establishes Ca^2+^ entry into mitochondria [54]. This mechanism has also been found to be important in human cancer [55,56]. MICU2 can interact with MICU1 and elevate the Ca^2+^ threshold activated by the MCU. Therefore, MICU2 can inhibit MCU activity at low Ca^2+^ concentrations [57].

Although MICU1, MICU2, and MICU3 belong to the same family, they have different effects on the MCU. MICU2 is the gatekeeper of the MCU, while MICU3 is an MCU activation enhancer. Overexpression of MICU3 causes a 10-fold increase in transient Ca^2+^ [58]. MCUb directly interacts with the MCU and mainly performs negative regulation of the MCU [59]. At present, the research results on the effect of the MCUR1 on the MCU are still controversial. Some studies have pointed out that the MCUR1, as an essential scaffold factor of the MCU complex [60], is the key component of the MCU complex. It has also been reported that mitochondrial Ca^2+^ uptake does not depend on the MCUR1, which is only a regulator that sets the Ca^2+^ threshold of the transition in mitochondrial permeability. Inhibiting the expression of the MCUR1 increases the Ca^2+^ threshold for inducing MPTP conversion, which can reduce the mitochondrial cell death that is induced by an overload of Ca^2+^ [61].

There is a sodium calcium transporter NCLX in the inner membrane of mitochondria, which is a sodium ion (Na^+^)-dependent Na^+^−Ca^2+^ reverse exchange channel and can positively regulate the outflow of Ca^2+^ in mitochondria [62]. When the concentration of Ca^2+^ in mitochondria is too high, it will enhance the activity of the NCLX and cause the opening of the MPTP on the inner membrane of mitochondria. Ca^2+^ is the center that regulates the MPTP. It can directly regulate the MPTP itself and indirectly affect the MPTP by regulating the adenosine diphosphate (ADP)/ATP balance, mitochondrial membrane potential, and ROS/reactive nitrogen level [63]. The study found that the MPTP has an important property: the increase in ADP and the recovery of Mg^2+^/Ca^2+^ caused by MPTP opening are reversible [64]. This reversibility makes MPTP opening have two modes: continuous opening and instantaneous opening, which can start the cell death signal pathway or maintain the normal physiological function of cells. In addition, there is LETM1 in the mitochondrial inner membrane [65]. When the concentration of Ca^2+^ in the mitochondrial matrix is low, LETM1 can transport Ca^2+^ into the matrix. On the contrary, Ca^2+^ is transported out of mitochondria. The study also found that silencing LETM1, despite the presence of the MCU, can still inhibit the influx of Ca^2+^ into mitochondria (Figure 2).

Mitochondrial Ca^2+^ homeostasis is unbalanced in tumors because in tumor cells, the cellular microenvironment is remodeled and leads to further mitochondrial Ca^2+^ imbalance, which is an adaptive phenomenon of tumors, and the mitochondrial Ca^2+^ imbalance will further promote the development of tumors. Some studies have shown that cancer cells can change mitochondrial Ca^2+^ homeostasis mainly through the following methods: (1) Ca^2+^ exists in a domain formed between the ER and the mitochondria, which is called the mitochondrial-associated membrane (MAM) and controls mitochondrial Ca^2+^ homeostasis [66] (Figure 2). Cancer cells can remodel their MAMs to affect mitochondrial Ca^2+^ homeostasis and promote cell survival, migration, invasion, metastasis, autophagy, and inhibit apoptosis [67,68,69]. (2) Mechano- and proton-sensing proteins may cause an imbalance in Ca^2+^ levels in cancer cells [70]. (3) In cancer cells, the expression and function of the magnesium (Mg^2+^) transporter are abnormal. The imbalance of Mg^2+^ homeostasis may destroy Ca^2+^ homeostasis [71]. (4) Cancer cells modify the Ca^2+^ signaling network by changing the expression and function of cation channels, pumps, or transporters [72].

## 3. Mitochondrial Ca^2+^ and Energy Metabolism of Tumor Cells

Ca^2+^ participates in almost all physiological activities in cells. Mitochondria were originally considered to be a “Ca^2+^ pool” with the ability to absorb a large amount of Ca^2+^, and the uptake of Ca^2+^ by mitochondria increases significantly when the extramitochondrial Ca^2+^ is overloaded [73]. It has been found that Ca^2+^ can stimulate glycogen decomposition and glucose oxidation, resulting in an increase in ATP supply [74]. The increase in cytoplasmic Ca^2+^ concentration is transmitted to mitochondria and Ca^2+^-activated dehydrogenase is a key rate control enzyme in the tricarboxylic acid cycle (TAC) flux. Ca^2+^ activation will lead to the increase in pyridine nucleotide reduction and oxidative phosphorylation [75]. Mitochondrial Ca^2+^ uptake can activate matrix enzymes, stimulate ATP production, and regulate energy metabolism by activating pyruvate dehydrogenase, isocitrate dehydrogenase, and ketoglutarate dehydrogenase. This “parallel activation model” provides a mechanism in which Ca^2+^ stimulates the process of energy consumption caused by physiological activities such as various hormones, muscle contraction, or increased cardiac load [76]. It also provides a means for cells to upregulate ATP supply to keep up with this energy consumption.

Metabolic reprogramming in tumor cells is considered to be a sign of cancer and is involved in tumor growth and development. Compared to normal cells, under the condition of sufficient aerobic supply, tumor cells still obtain energy by aerobic glycolysis and produce a large amount of lactic acid and a small amount of ATP [77]. M2 isoform of pyruvate kinase (PKM2) is critical for the metabolic fate of the glycolytic intermediates [78,79,80]. During the course of the disease, tumor cells will develop overall metabolic adaptability so that they can survive in the tumor microenvironment with low oxygen and nutrient levels [81].

In summary, Ca^2+^ affects the functional changes of mitochondria (such as mitochondrial dysfunction, metabolic conversion to glycolysis, and mtDNA mutations) and thus, cell energy metabolism, which is closely related to the occurrence and development of tumors (Figure 3). At present, the adaptability of tumor cell metabolism is the main limitation of cancer treatment, which is highly related to the resistance to therapeutic drugs [82]. The unique metabolic pattern of tumor cells is both a challenge and an opportunity. Understanding the metabolic mechanism of tumor cells is greatly significant for the early diagnosis of a tumor’s metabolic phenotype and rational targeted therapy.

## 4. Mitochondrial Ca^2+^ and the MCU in Autophagy/Mitophagy of Tumor Cells

Metabolic adaptations allow tumor cells to survive in the low oxygen and nutrient tumor microenvironment. Among these metabolic adaptations, tumor cells use glycolysis but also mitochondrial oxidation to generate ATP; another particular adaptation of tumor cell metabolism is the use of autophagy and mitophagy [83]. Autophagy plays a key role in maintaining cellular homeostasis [84]. Thus, autophagy disorders disrupt normal physiological processes and are implicated in the pathogenesis of various diseases, including tumors [85]. Autophagy is a highly conserved catabolic process that results in the degradation and recycling of proteins and organelles after the fusion of isolated vesicles, autophagosomes, and lysosomes that provide hydrolases [86]. The molecular process of autophagy is complex and involves sequential steps for nucleation, extension, and fusion of associated proteins, including autophagy-associated proteins [87]. Autophagy has two main physiological roles: the breakdown of dysfunctional proteins or organelles as a quality control mechanism and the recovery of biological macromolecules to maintain metabolic needs under nutritional stress [88]. Autophagy has been found to play two roles in a tumor: a protective role in the early stages of the tumor and the promotion of tumor growth in advanced stages [89].

Intracellular Ca^2+^ is considered a bidirectional regulator of autophagy [90,91], which may depend on the spatiotemporal parameters of Ca^2+^ signal transduction, nutrients, and the utilization of growth factors [92]. Ca^2+^ overload can affect autophagy, leading to normal cell carcinogenesis and the growth of tumor cells. It is demonstrated that Ca^2+^ agonists, such as vitamin D3 compounds, ionomycin, ATP, and thapsigargin, can stimulate the autophagy of MCF-7 breast tumor cells through Ca^2+^-activated kinase CaMKK [93]. Consistent with the activation of autophagy by Ca^2+^, researchers have found that mitochondrial fission-mediated Ca^2+^ signaling also significantly induces autophagy in HCC [94]. Conversely, some other research groups have found the inhibitory effect of Ca^2+^ on autophagy. At present, there are several ways for Ca^2+^ to inhibit autophagy: (1) the inositol 1,4,5-trisphosphate receptor (IP_3_R) mediates Ca^2+^ to reduce the release of Beclin1 so as to reduce autophagosome production and inhibit autophagy; (2) IP_3_R mediates Ca^2+^ activation of calpain, separates autophagy protein 5 from autophagy protein 12, reduces the level of their complex, and inhibits autophagy [95]; (3) The increase of Ca^2+^ released by the ER to the mitochondria enhances the TAC and ATP production and inhibits autophagy [96,97]; (4) IP_3_R mediates Ca^2+^ into mitochondria, resulting in increased ATP production and the inhibition of AMPK, thereby inhibiting autophagy. Therefore, Ca^2+^ may have different regulatory effects on autophagy.

As with non-selective autophagy, the role of mitophagy is complex and can depend on tumor type and stage. Since both autophagy and mitophagy are related to mitochondrial function, targeting mitochondrial ion channels may also be an interesting strategy to regulate autophagy or mitophagy in tumors. Ca^2+^ exchanges have been associated with autophagy and mitophagy regulation. Therefore, unsurprisingly, some mitochondrial calcium transporters, such as the MCU, have recently been found to be involved in autophagy and mitophagy regulation in tumor cells.

The MCU is generally considered to be the main Ca^2+^ transporter in the matrix, which is a major mediator of calcium influx into mitochondria. The MAM is an important part of Ca^2+^ transfer from the ER to the mitochondria to regulate mitochondrial enzymes. Ca^2+^ flow mainly occurs through IP_3_R and transient receptor potential cation channel subfamily M member 8 (TRPM8) in the ER membrane [98]. Sensitizing IP_3_Rs and the interruption of Ca^2+^ flow between the ER and the mitochondria break the calcium homeostasis and decrease mitochondrial bioenergetics, which subsequently decreases OXPHOS and activates autophagy [99,100]. However, unlike normal cells, autophagy activation caused by MAM destruction in tumor cells seems insufficient to maintain the required energy level, resulting in tumor cell death and reduced tumor growth [101] (Figure 4). Although the mechanisms linked with autophagy are not clearly understood, the MCU could be an interesting target to disrupt Ca^2+^ in the MAM in tumor cells, decreasing mitochondrial function and inducing cell death. The MCU has also been found to be altered in tumors from different tissues [102]. In particular, the expression of the MCU is associated with tumor progression and metastasis [103]. Therefore, mitochondrial Ca^2+^ and the MCU represent attractive antitumor targets for regulating mitochondrial dysfunction and autophagy/mitophagy in tumors.

## 5. Mitochondrial Ca^2+^ and Tumor Cell Apoptosis

Apoptosis involves the activation, expression, and regulation of a series of genes. It is not a phenomenon of autologous injury under pathological conditions, but a death process actively striving for better adaptation to the living environment [104]. The regulation of apoptosis is controlled by a very complex signal network system. There are three major signaling pathways: the mitochondrial pathway, the death receptor pathway, and the ER pathway [105]. These signal transduction pathways can eventually activate caspase-3, the executor of apoptosis, which hydrolyzes various cellular components and causes cell apoptosis [106]. In animal cells, the mitochondrial pathway is the most common apoptotic mechanism and the core of apoptosis [107,108]. In the early stage of apoptosis, mitochondria show changes such as increased permeability, Ca^2+^ uptake, decreased transmembrane potential, and the release of cytochrome C and apoptosis-inducing factors [109]. Changes in Ca^2+^ concentration may play a key role in the early apoptotic signal transduction pathway upstream of mitochondria [110]. However, this sensitive system can be affected to drive malignant transformation in cells.

In the process of apoptosis, intracellular Ca^2+^ overload can come from either extracellular Ca^2+^ influx or the release of the intracellular Ca^2+^ pool [111]. Some studies have suggested that the release of the intracellular Ca^2+^ pool can only cause a temporary increase in Ca^2+^, which is not enough to cause apoptosis. The triggering of apoptosis requires Ca^2+^ to reach a certain threshold and maintain this level for a long time [112]. With further research, there are many factors regulating Ca^2+^ levels in mitochondria, including intracellular regulation of the Bcl-2 family, the release of calcium pool ER and the participation of ROS [113]. At present, more than 20 members of the Bcl-2 family have been found. The proteins in the Bcl-2 family are widely distributed in the outer membrane of the mitochondria, nuclear membrane, and ER, regulating the activity of the caspases. Bcl-2 family members can be divided into three groups according to their structure and function. The first group includes Bcl-2, Bcl-XL, and Bcl-W, which have anti-apoptotic properties. The second group is a member of the Bcl-2 family with BH3-only proteins, which could increase the permeability of the mitochondrial outer membrane during cell apoptosis [114]. The third group, which contains all the domains except BH4, also increases membrane permeability and has pro-apoptotic activity [115].

The ER is an important Ca^2+^ reservoir in eukaryotic cells, so Ca^2+^ in the ER must maintain a stable level to ensure the accuracy of the Ca^2+^ signal [116]. Ca^2+^ released from the ER can directly flow into mitochondria and the uptake rate of mitochondrial Ca^2+^ depends on the concentration gradient of cytoplasmic Ca^2+^ at the IP_3_R opening on the ER. The opening of the ER InsP3/Ca^2+^ channel affects the Ca^2+^ balance in mitochondria and the InsP3/Ca^2+^ channel is one of the targets of caspase-3 [117]. Moreover, ER stress induced by the disturbance in the ER calcium state can activate caspase-12, a specific ER-localized protein, to trigger apoptosis in a mitochondria-independent way [118,119].

Mitochondria are the central link mediating apoptosis, as well as the main site of ROS generation [120]. With the discovery and further understanding of the role of mitochondrial Ca^2+^ in apoptosis, the research on the role of ROS in apoptosis is getting more and more in-depth. The regulation mechanisms of ROS on mitochondrial Ca^2+^ homeostasis are as follows: (1) After cells receive the pro-apoptotic signals, the increase of ROS promotes mitochondrial Ca^2+^ influx, which may be caused by affecting voltage-dependent Ca^2+^ channels, non-specific cell membrane Ca^2+^ permeability changes, and Na^+^/Ca^2+^ exchanges [121]; (2) Increased intracellular Ca^2+^ can activate other enzymes to further upregulate the level of oxygen free radicals, so ROS can indirectly produce more oxides and further promote the rise in mitochondrial Ca^2+^ level [122]. In addition, an overload of Ca^2+^ leading to oxidative metabolism impairment and ROS overproduction [123]. Previous studies have suggested that under oxygen stress, ROS produced by mitochondria will cause membrane lipid peroxidation and changes in mitochondrial function, resulting in the release of Ca^2+^ and apoptosis of mitochondria [124]; (3) ROS can regulate IP_3_R production and affect Ca^2+^ release from the ER into mitochondria [125]; (4) ROS can also affect the sarcoplasmic reticulum Ca^2+^ pump and inhibit intracellular or extracellular ER Ca^2+^ transfer by inhibiting the Ca^2+^-ATPase pump [126]; (5) Both ROS and Ca^2+^ can induce MPTP opening. On the other hand, MPTP opening leads to a large increase in ROS [127].

Several types of tumor cells have experienced extensive reorganization of their internal Ca^2+^ signal transduction mechanism, which promotes the occurrence of tumors [128]. Calcium ion exchange between mitochondria and the ER can be carried out through some Ca^2+^ signal proteins, including VDAC1, IP_3_R, and SERCA, which play vital roles in the processes of tumors. VDAC1 plays a significant role in cellular Ca^2+^ homeostasis and it has also been recognized as a key protein in mitochondria-mediated apoptosis [129]. For example, in several types of non-small cell lung cancer and cervical cancer, the expression level of VDAC1 is related to tumor growth and invasion [130]. The downregulation of IP_3_R1 in bladder cancer cells prevents mitochondrial Ca^2+^ overload by decreasing the uptake of ER−mitochondria Ca^2+^, thereby reducing cisplatin-mediated apoptosis [131]. The significant reduction or loss of SERCA3 subtypes in transformed colonic epithelial cells also proves that the Ca^2+^ signal is remodeled in tumorigenesis [132].

Recently, it has been found that in several cancer types, the imbalance of two new mechanisms will affect the renewal of the proteasome, so as to regulate the apoptosis sensitivity of tumor cells by affecting IP_3_R3 proteins and interfering with the Ca^2+^ exchange between the ER and mitochondria [133]. (1) The tumor suppressor protein PTEN and F-box/LRR repeat protein 2 (FBXL-2) compete for binding to IP_3_R3, which slows down FBXL-2-mediated IP_3_R3 proteasome degradation. This represents a new mechanism. The deletion of PTEN enables tumor cells to avoid apoptosis [134]. The downregulation of IP_3_R3 impairs the pro-apoptotic mitochondrial Ca^2+^ transfer. (2) The tumor suppressor protein BRCA1-associated protein 1 (BaP1) is a deubiquitinase that promotes the transfer of ER−mitochondria Ca^2+^ by stabilizing IP_3_R3. Under long-term environmental pressure, the function of BaP1 will be seriously disrupted, which is related to the acquired inactivating mutations of the BaP1 gene. The loss of BaP1 will lead to the downregulation of IP_3_R3, which hinders the effective apoptotic clearance of damaged cells and is conducive to the occurrence of tumors and the survival of malignant cells [135]. In addition, oncogenes and tumor suppressor proteins can play other roles in cancer development through Ca^2+^ signal regulation, such as resistance to apoptosis. Because mitochondrial Ca^2+^ overload is related to apoptosis and death, modifying ER−mitochondria Ca^2+^ transfer at the MAM will change the sensitivity of apoptosis, and tumor cells can acquire resistance to cell death accordingly [136]. For example, by inhibiting IP_3_R-mediated Ca^2+^ signaling or increasing the transmembrane distance at the MAM, the efficiency of ER−mitochondria Ca^2+^ transfer can be reduced, so as to decrease the sensitivity of tumor cells to apoptosis [137] (Figure 4).

## 6. The Relationship between the MCU and the Tumor

With the deepening of the research on the mechanism of cancer metastasis, the relationship between mitochondrial calcium homeostasis and the development of malignant tumors has attracted much attention [138,139]. The MCU is a major mediator of calcium influx into mitochondria and controls cellular energy metabolism, autophagy/mitophagy, and apoptosis. In most cancer tissues, the MCU showed moderate to strong immunostaining [140]. Increasingly, evidence shows that the MCU is closely related to multiple cancers, such as breast cancer, HCC, and colon cancer.

The MCU plays an important role in controlling the energy metabolism of tumor cells. The receptor-interacting protein kinase 1 (RIPK1) is an important signal molecule in the pathway of cell survival, apoptosis, and necrosis, which is significantly upregulated in colorectal cancer (CRC) cells. RIPK1 interacts with the MCU to promote CRC cell proliferation by increasing mitochondrial Ca^2+^ uptake and energy metabolism [141]. Compared with normal tissues, the MCU, MICU1, and MICU2 were overexpressed in oral squamous cell carcinoma (OSCC) tissues. The MCU is a new proto-oncogene of OSCC, which is regulated by nuclear factor erythroid 2-related factor 2 (Nrf2) transcription. The MCU can enhance the proliferation of OSCC cells and inhibit apoptosis [142]. Dihydroartemisinin can repress the proliferation and migration of OSCC cells by inhibiting the expression of the MCU [143]. MCU-mediated high mitochondrial Ca^2+^ can increase the proliferation of prostate cancer cells by inhibiting MPTP [144]. The MCU is involved in the autophagy of cancer cells. In kidney cancer cells, the upregulation of miR501-5p leads to the downregulation of the MCU, which leads to the activation of AMPK, thus promoting mTOR-independent autophagy [145]. The MCU also affects the apoptosis of cancer cells. Cathepsin S (CTSS) is overexpressed in glioblastomas (GBs). High levels of CTSS are associated with tumor progression and a poor prognosis of GB. Inhibiting the expression of CTSS in GB cells can increase the expression of MCUs. Enhanced mitochondrial Ca^2+^ uptake leads to mitochondrial Ca^2+^ overload, produces a large number of ROS, and, finally, causes apoptosis [146]. RY10-4 can induce the apoptosis of breast cancer cells by elevating Ca^2+^ through the MCU [147]. The MCUR1 is frequently upregulated in HCC cells, which enhances Ca^2+^ uptake into mitochondria in an MCU-dependent manner. The HCC cell survival rate is significantly improved by inhibiting mitochondrial-dependent apoptosis and promoting HCC cell proliferation, resulting in poor prognosis [148]. The data also show that miR-25 decreases mitochondrial Ca^2+^ uptake through selective MCU downregulation, thereby reducing apoptosis. The MCU seems to be downregulated in human colon cancer samples. Correspondingly, miR-25 is abnormally expressed, indicating that mitochondrial Ca^2+^ plays an important role in the survival of cancer cells [149].

Current studies have suggested that the MCU is correlated with tumor size and lymphatic infiltration, which may contribute to tumor growth and metastasis [150,151]. It is speculated that the MCU affects the expression of VEGF through HIF-1α and the inhibition of MCU expression significantly reduces the invasion and migration ability of breast cancer cells [24,152]. In addition, the expression of the MCUR1 significantly affects the progression and prognosis of breast cancer [153,154]. However, the role of the MCU in cancer research remains controversial. Studies have shown that specific Ca^2+^ channels play different roles in some cancers due to different regulatory mechanisms. Previous studies have revealed that a highly expressed MCU promotes the metastasis of adrenocortical carcinoma breast cancer cells with poor prognosis. In hepatocellular carcinoma studies, MCU-dependent mitochondrial Ca^2+^ uptake promotes metastasis of HCC cells [155]. In fact, we analyzed the transcriptional expression levels of MCUs in different cancers through the relevant database (https://tnmplot.com/analysis/; accessed on 31 March 2022) [156] and demonstrated that the expression levels of MCUs in most tumors are not consistent with those in normal tissues (Figure 5). The majority of tumors have significantly elevated levels of MCU expression. However, high MCU expression in cancer patients may not always be beneficial. Coincidentally, we analyzed the survival curves between MCU expression levels and cancer patient survival through the GEPIA database (http://gepia.cancer-pku.cn/index.html; accessed on 31 March 2022) and found that in adrenocortical carcinoma and hepatocellular carcinoma, overall survival is significantly greater in low MCU expression than in high MCU expression (Figure 6A,B). On the contrary, in renal clear cell carcinoma and brain lower grade glioma, overall survival is significantly greater in high MCU expression than in low MCU expression (Figure 6C,D). To sum up, the relationship between the MCU and tumor is complex and needs more in-depth research.

Mitochondria regulate Ca^2+^ homeostasis through the uptake of Ca^2+^ into the mitochondria via MCU and the release of Ca^2+^ from the mitochondria via NCLX, regulating intramitochondrial and intracytoplasmic Ca^2+^ concentrations. Therefore, the regulation of both is deeply intertwined. Since the reorganization of cytosolic calcium signaling commonly occurs in tumor cells, mitochondrial calcium imbalance causes alterations in cytosolic calcium signaling and thus, affects tumorigenesis and progression [157]. Given the important impact of mitochondrial calcium imbalance on tumors, a large number of studies have used mitochondrial calcium imbalance as a starting point to explore new diagnostic and therapeutic approaches to tumors. It is found that proteins associated with mitochondrial calcium uptake may serve as novel biomarkers for predicting poor prognosis in HCC. This study includes tumor specimens and adjacent normal liver tissue from 354 patients with confirmed HCC as study subjects and concluded that HCC patients with low MICU1 and high MCU/MICU2 expression exhibited poor survival rates, overall survival rates and disease-free survival rates [158]. Another study shows that the MCUR1 promotes in vitro invasion and in vivo metastasis of HCC cells by promoting epithelial−mesenchymal transition. This process is mainly done by the MCUR1 through the activation of the ROS/Nrf2/Notch1 pathway. It has also been found that treatment with the mitochondrial Ca^2+^-buffering protein parvalbumin significantly inhibits the ROS/Nrf2/Notch pathway, MCUR1-induced epithelial− mesenchymal transition and HCC metastasis [159]. In a study of CRC, the MCU is markedly increased in CRC tissues, and upregulated MCU is associated with poor prognosis in patients with CRC [160]. An upregulated MCU enhances mitochondrial Ca^2+^ uptake and causes mitochondrial Ca^2+^ imbalance, which, in turn, promotes CRC cell growth in vitro and in vivo. Ru360 is a highly potent and selective MCU inhibitor that can effectively block MCU-mediated mitochondrial Ca^2+^ uptake and, ultimately, slow CRC progress. These results may provide a potential pharmacological target for CRC treatment [161]. Saverio Marchi’s group demonstrated for the first time that MCUs are suitable targets for miRNA-25, which reduces prostate and colon cancer cells’ dependence on Ca^2+^ [162]. Therefore, we can induce the apoptosis in cancer cells by reducing MCU protein levels and mitochondrial Ca^2+^ uptake.

## 7. Conclusions

The occurrence and development of tumors is a complex process regulated by multiple signaling networks. In this paper, we summarize, analyze, and discuss that mitochondrial Ca^2+^ and the MCU play crucial roles in energy metabolism, autophagy/mitophagy, and apoptosis of tumor cells. The discovery of the MCU and its regulatory proteins represents a new era of research on MCU-mediated mitochondrial Ca^2+^ dyshomeostasis in cancer. Currently, drug candidates targeting the MCU or its regulatory factors are still emerging. Although a flurry of studies has confirmed the correlation between mitochondrial Ca^2+^ dyshomeostasis and the progression of a variety of tumors, the exact mechanism and targeted therapy remain to be further elucidated. The tumor diagnosis and treatment strategy for mitochondrial Ca^2+^ homeostasis will bring a new dawn to tumor risk prediction, precancerous lesion screening, clinical targeted therapy, and prognosis assessment.

## Figures and Tables

**Figure 1 ijms-23-06667-f001:**
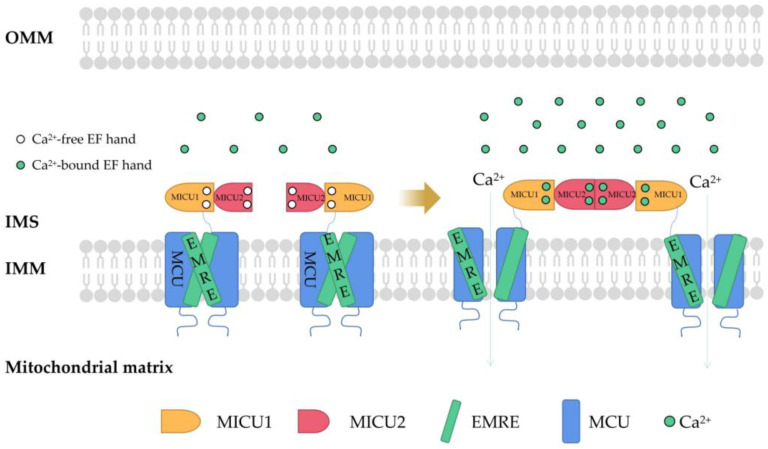
The structure of MCU and connections to its regulators. Mitochondrial Ca^2+^ uptake through MCU. In mammals, MCU contains four core components: pore-forming MCU protein, the gatekeepers MICU1 and MICU2, and an auxiliary subunit EMRE. MCU plays a vital role in Ca^2+^ transport. In order to prevent Ca^2+^ overload, the activity of MCU must be strictly regulated by MICUs, which can sense the change in cytosolic Ca^2+^ concentration to open and close the MCU. MCU, mitochondrial calcium uniporter; MICU, mitochondrial Ca^2+^ uptake; EMRE, essential MCU regulator; OMM, outer mitochondrial membrane; IMS, intermembrane space; IMM, inner mitochondrial membrane; EF hand, helix−loop−helix structure.

**Figure 2 ijms-23-06667-f002:**
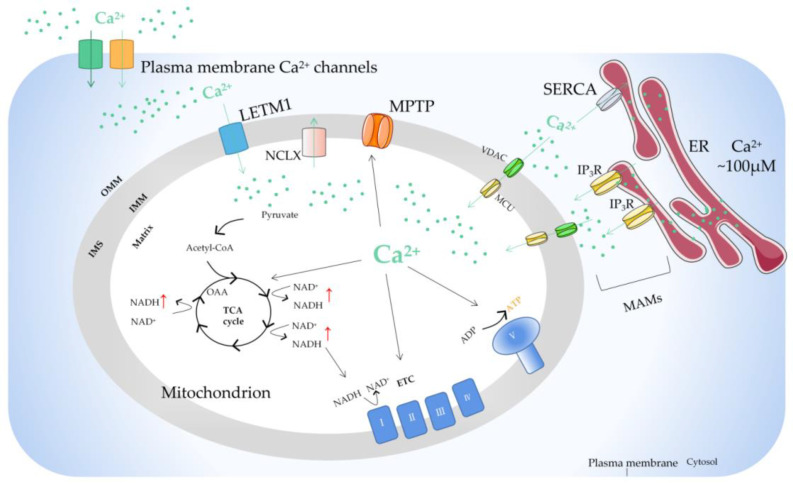
The basic mechanism of mitochondrial Ca^2+^ regulation. Ca^2+^ transfer from ER to mitochondria occurs on the MAMs, where there are special Ca^2+^ channels. The opening of IP3R on the surface of ER results in the release of Ca^2+^ from the lumen of ER. Ca^2+^ passes through OMM via VDAC and traverses IMM via MCU. Stimulus acts by producing Ca^2+^ mobilization signals, triggering the increase of intracellular Ca^2+^ concentration. The function of mitochondrial Ca^2+^ uptake and release are mainly to regulate the matrix Ca^2+^ level, thus regulating the activity of mitochondrial dehydrogenase, resulting in increased NADH and ATP production. Ca^2+^ can also activate mitochondrial ETC complexes. In the steady state, Ca^2+^ entering mitochondria through MCU must exit through one of the mitochondrial Ca^2+^ efflux mechanisms. ER, endoplasmic reticulum; MAM, mitochondrial-associated ER membrane; IP3R, inositol triphosphate receptor; MCU, mitochondrial calcium uniporter; VDAC, voltage-dependent anion-selective channel; ETC, electron transport chain; OMM, outer mitochondrial membrane; IMS, intermembrane space; IMM, inner mitochondrial membrane; LETM1, leucine zipper/EF hand-containing transmembrane-1; MPTP, mitochondrial permeability transition pore; NCLX, mitochondrial Na^+^/Ca^2+^ exchanger; SERCA, sarco-endoplasmic reticulum Ca^2+^-ATPase; OAA, oxaloacetic acid; TCA cycle, tricarboxylic acid cycle.

**Figure 3 ijms-23-06667-f003:**
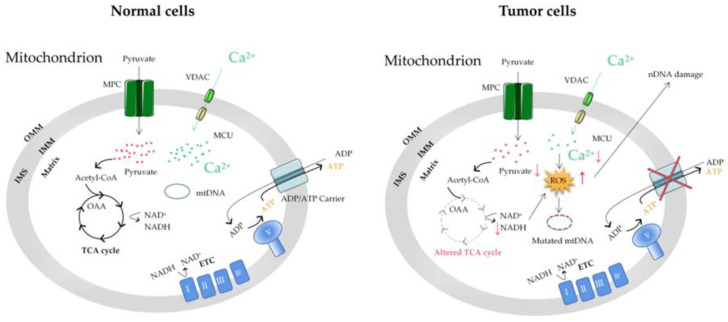
The mitochondrial Ca^2+^ and energy metabolism in normal and tumor cells. The reprogramming of energy metabolism, including energy production disorders caused by cell respiratory defects, is the core symbol of cancer. The change in energy metabolism in cancer cells is related to the abnormal function of mitochondria. Accumulation of the ROS induced by mitochondrial Ca^2+^ dyshomeostasis and altered TCA in tumor cells can cause mitochondrial DNA (mtDNA) and nuclear DNA (nDNA) mutations. MCU, mitochondrial calcium uniporter; VDAC, voltage-dependent anion-selective channel; ETC, electron transport chain; OMM, outer mitochondrial membrane; IMS, intermembrane space; IMM, inner mitochondrial membrane; MPC, mitochondrial pyruvate carrier; OAA, oxaloacetic acid; TCA cycle, tricarboxylic acid cycle.

**Figure 4 ijms-23-06667-f004:**
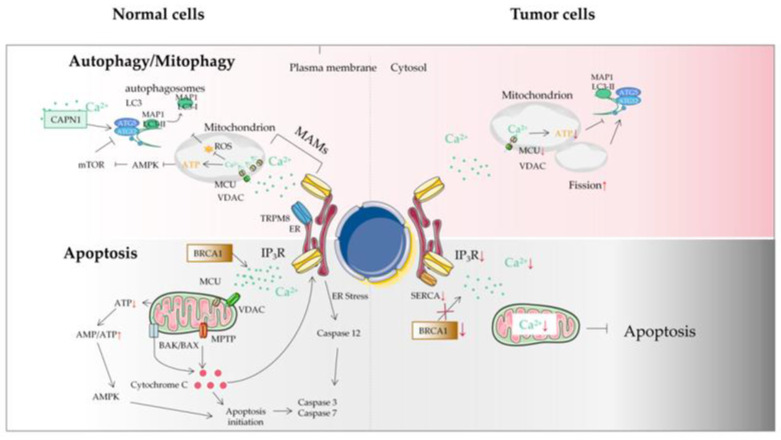
The autophagy/mitophagy and apoptosis of tumor cells. Autophagy plays a key role in maintaining cellular homeostasis. Autophagy disorder destroys normal physiological processes and can lead to cancer. Ca^2+^ can inhibit autophagy through an IP3R- or ER-mediated manner. Some mitochondrial Ca^2+^ transporters are also involved in autophagy and mitophagy regulation. Autophagy plays two roles in a tumor: a protective role in the early stages of tumor and the promotion of tumor growth in advanced stages. Tumor cells may avoid apoptosis by reducing Ca^2+^ influx into the cytoplasm. It can be achieved by downregulation of the expression of Ca^2+^ channels in the plasma membrane or by reducing the effectiveness of the signal pathways that activate these channels. This protective measure will greatly reduce the response of Ca^2+^ overload to pro-apoptotic stimulus, thus impairing the effectiveness of mitochondrial and cytoplasmic apoptotic pathways in tumor cells. Another mechanism is that tumor cells adapt to the reduction of Ca^2+^ in ER, without inducing the pro-apoptotic ER stress response usually accompanied by ER Ca^2+^ imbalance. ER, endoplasmic reticulum; MAM, mitochondrial-associated ER membrane; MCU, mitochondrial calcium uniporter; MPTP, mitochondrial permeability transition pore; ROS, reactive oxygen species; TRPM8, transient receptor potential cation channel subfamily M member 8; VDAC, voltage-dependent anion-selective channel; SERCA, sarco-endoplasmic reticulum Ca^2+^-ATPase; IP3R, inositol triphosphate receptor; BRCA1, breast cancer susceptibility gene.

**Figure 5 ijms-23-06667-f005:**
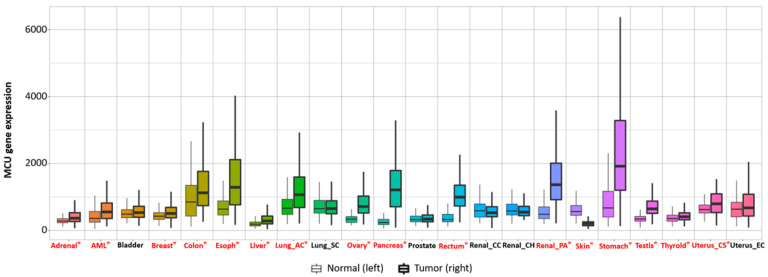
Transcriptional expression level of the MCU in various normal and cancerous organs. The MCU is closely related to multiple cancers; the expression levels of the MCU in most tumors are not consistent with those in normal tissues. The majority of tumors have significantly elevated levels of MCU expression. Significant differences by Mann−Whitney U test are marked with red and *. MCU, mitochondrial calcium uniporter.

**Figure 6 ijms-23-06667-f006:**
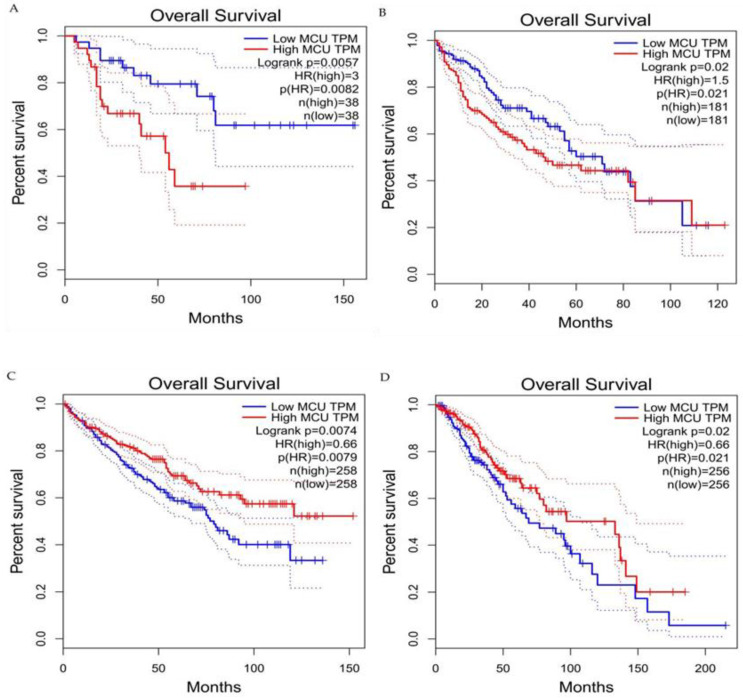
Survival curves for overall survival of high versus low expressing MCU. (**A**) Adrenocortical carcinoma. (**B**) Hepatocellular carcinoma. (**C**) Renal clear cell carcinoma. (**D**) Brain lower grade glioma. In adrenocortical carcinoma and hepatocellular carcinoma, overall survival is significantly greater in low MCU expression than in high MCU expression. In renal clear cell carcinoma and brain lower grade glioma, overall survival is significantly greater in high MCU expression than in low MCU expression. MCU, mitochondrial calcium uniporter; HR, hazard rate.

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
