# Peer review of "The Regulatory Roles of Mitochondrial Calcium and the Mitochondrial Calcium Uniporter in Tumor Cells"

_ijms, 2022, doi:10.3390/ijms23126667_

Round 1

Reviewer 1 Report

This is an interesting review of important aspects concerning various aspects   how mitochondrial calcium homeostasis is regulated in tumor cells, but influential and important aspects of this topic are missing, e.g. the work of Natalia Preverskaya concerning the regulation of calcium signaling in cancer  as well as the relevant contributions by the Lab of Geert Bultynck dealing with important aspects of autophagy and cancer. The significance of the data of Figs. 2 and 3 are difficult to evaluate since they are not obtained from a cited reference and therefore their value is difficult to be judged.

Refs. 115 and 116 are missing

Author Response

Response to Reviewer 1 Comments

Point 1: This is an interesting review of important aspects concerning various aspects how mitochondrial calcium homeostasis is regulated in tumor cells, but influential and important aspects of this topic are missing, e.g. the work of Natalia Preverskaya concerning the regulation of calcium signaling in cancer as well as the relevant contributions by the Lab of Geert Bultynck dealing with important aspects of autophagy and cancer. The significance of the data of Figs. 2 and 3 are difficult to evaluate since they are not obtained from a cited reference and therefore their value is difficult to be judged.

RESPONSE 1: We sincerely thank the reviewer for insightful suggestions that allow us to improve our manuscript. We have added a paragraph about the work of Natalia Preverskaya in “5. Mitochondrial Ca2+ and tumor cell apoptosis” section. Please see page 8 of the revised manuscript, lines 356-358. The changes in calcium ions often have influence on tumor progression. Calcium is also an important regulator of apoptosis, from initial to final stage of apoptotic bodies.

The findings of Geert Bultynck concerning autophagy and cancer were added to “4. Mitochondrial Ca2+ and autophagy of tumor cells” section. Please see page 5 of the revised manuscript, lines 254-255; page 6, lines 264-270; and page 6, lines 288-290. Calcium signals have bidirectional effects on autophagy. IP3Rs can transfer Ca2+ to the mitochondria and are involved in both autophagy and apoptosis.

Figs. 2 and 3 (Figs. 3 and 4 in the revised version) results were analyzed and discovered from public databases to illustrate the potentially important role of MCU in tumors. We believe these data play a crucial role in attracting and stimulating reader interest.

Point 2: Refs. 115 and 116 are missing

RESPONSE 2: Thanks and we have added additional references.

Reviewer 2 Report

This is an interesting review which deals with a current and evolving topic. Whilst there are a few areas where the manuscript can be improved with more careful wording and the addition of key/seminal original references a major issue in reviewing the manuscript is that the reference list is out of synch with the text citations.  It would be easier to review this manuscript if this was corrected since it makes checking references, citations and relevance difficult.

However, there are some other points that could be addressed at the same time.

Since the topic is mitochondrial Ca2+ homeostasis  the authors appear not to have included some key references, notably involving the first measurements  of mitochondrial Ca2+, the linking to of mitochondrial Ca2+ to Ca2+  release from intracellular stores and  then linking of mitochondrial  Ca2+ to  ATP production. There are key original papers by Rizzuto, colleagues  and others not cited although some reviews are acknowledged.    

These include:

Robb-Gaspers LD, Burnett P, Rutter GA, Denton RM, Rizzuto R, Thomas AP. Integrating cytosolic calcium signals into mitochondrial metabolic responses. EMBO J. 1998 Sep 1;17(17):4987–5000.

Jouaville LS, Pinton P, Bastianutto C, Rutter GA, Rizzuto R. Regulation of mitochondrial ATP synthesis by calcium: evidence for a long-term metabolic priming. Proc Natl Acad Sci U S A. 1999 Nov 23;96(24):13807–13812.

Hajnóczky G, Robb-Gaspers LD, Seitz MB, Thomas AP. Decoding of cytosolic calcium oscillations in the mitochondria. Cell. 1995 Aug 11;82(3):415–424.

RIZZUTO R, BRINI M, MURGIA M, POZZAN T (1993) Microdomains with high Ca2+ close to IP3-sensitive channels that are sensed by neighboring mitochondria. Science 262:744-747

Diego De Stefani, Anna Raffaello, Enrico Teardo, Ildikò Szabò, Rosario Rizzuto.  A forty-kilodalton protein of the inner membrane is the mitochondrial calcium uniporter 2011 Jun 19;476(7360):336-40. doi: 10.1038/nature10230 Integrative genomics identifies MCU as an essential component of the mitochondrial calcium uniporter.

Baughman JM, Perocchi F, Girgis HS, Plovanich M, Belcher-Timme CA, Sancak Y, Bao XR, Strittmatter L, Goldberger O, Bogorad RL, Koteliansky V, Mootha VK. Nature. 2011 Jun 19;476(7360):341-5. doi: 10.1038/nature10234.

There are some more specific comments:

Line 13:  I would hesitate to call mitochondria a Ca2+ storage organelle. Yes, they take up and release Ca2+ but they certainly don’t act like a functional store such as the ER or golgi.  Avoid the double negative ‘ non-negliable’.

Line 23:  I think stating that this review enables clinical diagnosis and therapies is overreach. The review provides insight into how understanding mitochondrial Ca2+ regulation may aid diagnosis and therapy.  

Line 28: ‘large amount’ unnecessary comment. 

Line 31:  as with line 13.

Line32: intermembrane gap

Line 35 and 36: contraction used on both side of a conjunction

Line 38: be more specific about the Ca2+ concentration

Line 63. Is the citation for 21 correct?

Lines 76 and 77. There are two… This sentence does not make sense.  The ER influences mitochondrial Ca2+ and so does cytoplasmic Ca2+ and even Ca2+ influx. However, no matter the source, the Ca2+ always transits via the cytoplasmic, all be it very locally in some cases.  Also, mitochondria don’t absorb Ca2+.  This implies the process is passive.

Line 78 and 79. Surely the reference should include De Stefani and Baughman (see list above).

Line 132: Reference 45 is not cited correctly in the list. The references are clearly out of sequence here.

Reference 46 is Denton (Ref 48)

Line 232: Ref 78 should be 76  in the list.

Manuscript has 114 references the list has 116.

The authors should reflect on the use of more  primary sources when describing key developments in the field. 

Author Response

Response to Reviewer 2 Comments

Point 1: This is an interesting review which deals with a current and evolving topic. Whilst there are a few areas where the manuscript can be improved with more careful wording and the addition of key/seminal original references a major issue in reviewing the manuscript is that the reference list is out of synch with the text citations. It would be easier to review this manuscript if this was corrected since it makes checking references, citations and relevance difficult.

RESPONSE 1: Thank you and we have revised.

Point 2: However, there are some other points that could be addressed at the same time.

Since the topic is mitochondrial Ca2+ homeostasis the authors appear not to have included some key references, notably involving the first measurements of mitochondrial Ca2+, the linking to of mitochondrial Ca2+ to Ca2+ release from intracellular stores and then linking of mitochondrial Ca2+ to ATP production. There are key original papers by Rizzuto, colleagues and others not cited although some reviews are acknowledged.    

These include:

Robb-Gaspers LD, Burnett P, Rutter GA, Denton RM, Rizzuto R, Thomas AP. Integrating cytosolic calcium signals into mitochondrial metabolic responses. EMBO J. 1998 Sep 1;17(17):4987–5000.

Jouaville LS, Pinton P, Bastianutto C, Rutter GA, Rizzuto R. Regulation of mitochondrial ATP synthesis by calcium: evidence for a long-term metabolic priming. Proc Natl Acad Sci U S A. 1999 Nov 23;96(24):13807–13812.

Hajnóczky G, Robb-Gaspers LD, Seitz MB, Thomas AP. Decoding of cytosolic calcium oscillations in the mitochondria. Cell. 1995 Aug 11;82(3):415–424.

RIZZUTO R, BRINI M, MURGIA M, POZZAN T (1993) Microdomains with high Ca2+ close to IP3-sensitive channels that are sensed by neighboring mitochondria. Science 262:744-747

Diego De Stefani, Anna Raffaello, Enrico Teardo, Ildikò Szabò, Rosario Rizzuto.  A forty-kilodalton protein of the inner membrane is the mitochondrial calcium uniporter 2011 Jun 19;476(7360):336-40. doi: 10.1038/nature10230 Integrative genomics identifies MCU as an essential component of the mitochondrial calcium uniporter.

Baughman JM, Perocchi F, Girgis HS, Plovanich M, Belcher-Timme CA, Sancak Y, Bao XR, Strittmatter L, Goldberger O, Bogorad RL, Koteliansky V, Mootha VK. Nature. 2011 Jun 19;476(7360):341-5. doi: 10.1038/nature10234.

RESPONSE 2: Thank you for pointing out these points. We have added relevant descriptions according to the proposed original papers. Relevant descriptions were added  to “1. Introduction” section and “2. Regulation of mitochondrial Ca2+” section. Please see page 2 of the revised manuscript, lines 54-62; page 2, lines 64-66; and page 3, lines 122-126.

There are some more specific comments:

Point 3: Line 13:  I would hesitate to call mitochondria a Ca2+ storage organelle. Yes, they take up and release Ca2+ but they certainly don’t act like a functional store such as the ER or golgi.  Avoid the double negative ‘non-negliable’.

RESPONSE 3: We thank the reviewer for pointing out this issue, and we have changed the description more clearly in revised manuscript. Please see page 1 of the revised manuscript, lines 12-14.

Point 4: Line 23:  I think stating that this review enables clinical diagnosis and therapies is overreach. The review provides insight into how understanding mitochondrial Ca2+ regulation may aid diagnosis and therapy.  

RESPONSE 4:  We sincerely thank the reviewer for this insightful comment, and we have made corresponding changes. Please see page 1 of the revised manuscript, lines 21-25.

Point 5: Line 28: ‘large amount’ unnecessary comment.

RESPONSE 5: Thanks and we have revised. Please see page 1 of the revised manuscript, lines 30.

Point 6: Line 31:  as with line 13.

RESPONSE 6: Thanks and we have revised. Please see page 1 of the revised manuscript, lines 34-35.

Point 7: Line32: intermembrane gap

RESPONSE 7: Our apology and corrected. Please see page 1 of the revised manuscript, lines 35.

Point 8: Line 35 and 36: contraction used on both side of a conjunction

RESPONSE 8: Thanks and we have revised. Please see page 1 of the revised manuscript, lines 37-40.

Point 9: Line 38: be more specific about the Ca2+ concentration

RESPONSE 9: We thank the reviewer for this suggestion and we have added relevant discussion in “1. Introduction” section and “2. Regulation of mitochondrial Ca2+” section. Please see page 1 of the revised manuscript, lines 43-45; page 2, lines 46-48; page 2, lines 96-98; and page 3, lines 99-112.

Point 10: Line 63. Is the citation for 21 correct?

RESPONSE 10: Our apology and corrected. Please see page 2 of the revised manuscript, lines 87-88.

Point 11: Lines 76 and 77. There are two… This sentence does not make sense. The ER influences mitochondrial Ca2+ and so does cytoplasmic Ca2+and even Ca2+influx. However, no matter the source, the Ca2+ always transits via the cytoplasmic, all be it very locally in some cases.  Also, mitochondria don’t absorb Ca2+.  This implies the process is passive.

RESPONSE 11: Thanks and we have revised. Please see page 3 of the revised manuscript, lines 120-122.

Point 12: Line 78 and 79. Surely the reference should include De Stefani and Baughman (see list above).

RESPONSE 12: Thanks and we have added the relevant references. Please see page 3 of the revised manuscript, lines 122-126.

Point 13: Line 132: Reference 45 is not cited correctly in the list. The references are clearly out of sequence here.

RESPONSE 13: Thanks and we have revised. Please see page 4 of the revised manuscript, lines 190.

Point 14: Reference 46 is Denton (Ref 48)

RESPONSE 14: Thanks and we have revised. Please see page 4 of the revised manuscript, lines 198.

Point 15: Line 232: Ref 78 should be 76 in the list.

RESPONSE 15: Thanks and we have revised. Please see page 7 of the revised manuscript, lines 315.

Point 16: Manuscript has 114 references the list has 116.

RESPONSE 16: Thanks and we have revised.

Point 17: The authors should reflect on the use of more primary sources when describing key developments in the field.

RESPONSE 17: Thank you for this suggestion and we have revised. Please see page 3 of the revised manuscript, lines 105-112; page 6, lines 216-218; page 6, lines 267-268; page 8, lines 370-371.

Reviewer 3 Report

The present submission is a very timely review of the role of mitochondrial calcium homeostasis and regulation in tumor cells. The article is clearly written and the data presented are interesting. However, the content of this review appears very limited, and some essential studies and concepts are missing. Consequently, this review provides limited information on the situation and should be expanded and further elaborated. Below, some emerging novel aspects of mitochondrial Ca2+ in cancer are listed. The authors are encouraged to pick up these concepts and check the papers indicated (pmid…PubMed ID) and further elaborate this important review to a comprehensive but informative and more complete overview.

Major points:

1. It would be important to consider the impact of matrix Ca2+ on mitochondrial TCA cycle as essential aspect for boosting ATP production for cancer growth. However in this aspect, please consider the limited PKM2 activity and the revisit of the Warburg erffect (pmid: 20156581; pmid: 24747424; pmid: 33246010).

2. MICU1 has been shown to be methylated by proteine arginine methyltransferase 1 in cancer cells yielding decreased Ca2+ sensitivity and reduced Ca2+ entry, UCP2 that is essential for mitochjondrial Ca2+ uptake in cancer cells (pmid: 17351641) binds to methylated MICU! and normalizes the Ca2+ sensitivity of MICU1 aand reestablishes Ca2+ entry into mitochodnria (pmid: 27642082). This mechanism has been also found to be important in human cancer (pmid: 29113301; pmid: 31230909).

3. The tethering between mitochondria and the endoplasmic reticulum is altered in cancer, possibly a survival strategy of the cancer cells (pmid: 33614647). Changes thereof might kill cancer cells selectively (pmid: 27606689).

4. MICU1 also regulates cristae junction and, thus, affects the efficiency of the OXPHOS machinery (pmid: 31427612; pmid: 34915234).

5. MCUR1 is a marker for the progression and prognosis of breast cancer (doi: 10.21203/rs.3.rs-1552718/v1). 

6. Ca2+/Redox balance and disturbances of mitohormesis should be discussed as well (pmid: 33811561).

7. Please present more schemata to help the readers.

Minor points:

  • please use superscripts for the “2+” in Ca2+ also in the abstract

Author Response

Response to Reviewer 3 Comments

The present submission is a very timely review of the role of mitochondrial calcium homeostasis and regulation in tumor cells. The article is clearly written and the data presented are interesting. However, the content of this review appears very limited, and some essential studies and concepts are missing. Consequently, this review provides limited information on the situation and should be expanded and further elaborated. Below, some emerging novel aspects of mitochondrial Ca2+ in cancer are listed. The authors are encouraged to pick up these concepts and check the papers indicated (pmid…PubMed ID) and further elaborate this important review to a comprehensive but informative and more complete overview.

RESPONSE: We sincerely thank the reviewer for the encouraging and insightful comments and suggestions that allow us to improve our manuscript. We have made corresponding changes based on the valuable comments of the reviewer, and hope that the revised manuscript will provide a more informative and complete overview. 

Major points:

Point 1: It would be important to consider the impact of matrix Ca2+on mitochondrial TCA cycle as essential aspect for boosting ATP production for cancer growth. However, in this aspect, please consider the limited PKM2 activity and the revisit of the Warburg erffect (pmid: 20156581; pmid: 24747424; pmid: 33246010).

RESPONSE 1: Thanks for reviewer’s excellent advice, and we have added relevant description to the “3. Mitochondrial Ca2+ and energy metabolism of tumor cells” section. Please see page 5 of the revised manuscript, lines 214-218.

Point 2: MICU1 has been shown to be methylated by protein arginine methyltransferase 1 in cancer cells yielding decreased Ca2+sensitivity and reduced Ca2+entry, UCP2 that is essential for mitochondrial Ca2+ uptake in cancer cells (pmid: 17351641) binds to methylated MICU! and normalizes the Ca2+sensitivity of MICU1 and reestablishes Ca2+ entry into mitochondria (pmid: 27642082). This mechanism has been also found to be important in human cancer (pmid: 29113301; pmid: 31230909).

RESPONSE 2: Sincerest thanks for reviewer’s comments on our manuscript. We have carefully modified the paper in response to the extensive and insightful reviewer comments, we have added relevant description to the “2. Regulation of mitochondrial Ca2+” section. Please see page 4 of the revised manuscript, lines 157-162.

Point 3: The tethering between mitochondria and the endoplasmic reticulum is altered in cancer, possibly a survival strategy of the cancer cells (pmid: 33614647). Changes thereof might kill cancer cells selectively (pmid: 27606689).

RESPONSE 3: We thank the reviewer for this suggestion and we have added relevant discussion in “4. Mitochondrial Ca2+ and autophagy of tumor cells” section. Please see page 6 of the revised manuscript, lines 258-262.

Point 4: MICU1 also regulates cristae junction and, thus, affects the efficiency of the OXPHOS machinery (pmid: 31427612; pmid: 34915234).

RESPONSE 4: We thank the reviewer for this suggestion and we have added relevant discussion in “2. Regulation of mitochondrial Ca2+” section. Please see page 3 of the revised manuscript, lines 149-151.

Point 5: MCUR1 is a marker for the progression and prognosis of breast cancer (doi: 10.21203/rs.3.rs-1552718/v1).

RESPONSE 5: We thank the reviewer for this suggestion and we have added relevant discussion in “6. The relationship between MCU and tumor” section. Please see page 9 of the revised manuscript, lines 426-427.

Point 6: Ca2+/Redox balance and disturbances of mitohormesis should be discussed as well (pmid: 33811561).

RESPONSE 6: We thank the reviewer for this suggestion and we have added a paragraph about Ca2+ overload- and ROS-associated mitochondrial dysfunction in “5. Mitochondrial Ca2+ and tumor cell apoptosis” section. Please see page 8 of the revised manuscript, lines 370-371.

Point 7: Please present more schemata to help the readers.

RESPONSE 7: We thank the reviewer for this good suggestion, and have added Figure 1 in revised manuscript to help the readers.

Minor points:

Point 8: please use superscripts for the “2+” in Ca2+ also in the abstract

RESPONSE 8: Our apology and corrected.

Round 2

Reviewer 1 Report

The authors carefully followed the advice by the reviewer and changed the manuscript accordingly. The revised version of the manuscript is now acceptable for publication.

Author Response

The authors carefully followed the advice by the reviewer and changed the manuscript accordingly. The revised version of the manuscript is now acceptable for publication.

Response: Thank you very much for your encouraging and insightful comments.

Reviewer 3 Report

Thank you for addressing my remarks very well. I think this work is very interesting and worthy of publication in the IJMS.

Author Response

Thank you for addressing my remarks very well. I think this work is very interesting and worthy of publication in the IJMS.

Response: Thank you very much for your encouraging comments and insightful suggestions that allow us to improve our manuscript.